SciPost Physics

Submission

MPP-2024-236

# A tale of $Z$+jet: SMEFT effects and the Lam-Tung relation

R. Gauld,[1] U. Haisch,[1] and J. Weiss[1,2]

**1** Werner-Heisenberg-Institut, Max-Planck-Institut für Physik,
Boltzmannstraße 8, 85748 Garching, Germany

**2** Technische Universität München, Physik-Department,
James-Franck-Straße 1, 85748 Garching, Germany

rgauld@mpp.mpg.de, haisch@mpp.mpg.de, jweiss@mpp.mpg.de

December 18, 2024

## Abstract

We derive constraints on dimension-six light-quark dipole operators within the Standard Model (SM) effective field theory, based on measurements of $Z$ production at SLC and LEP, as well as $Z$+jet production at the LHC. Our new constraints exclude the parameter space that could potentially explain the observed discrepancy between theoretical predictions and experimental data for the Lam-Tung relation. With these updated limits, we model-independently determine the maximum possible influence that beyond-SM contributions could have on the angular coefficients $A_0$ and $A_2$, which enter the Lam-Tung relation.

# 1 Motivation

Precision measurements at the Large Hadron Collider (LHC) are key to testing the Standard Model (SM). Processes with a distinct and clean signature, such as the decay of a $Z$ boson into two charged leptons, are thereby of particular interest. In fact, the transverse momentum $(p_T)$ distribution of the $Z$ boson in neutral-current (NC) Drell-Yan (DY) production is one of the most precisely measured and predicted observables at the LHC, with an experimental accuracy of better than 1% for $p_{T,Z}$ values below $250\,\text{GeV}$ [1, 2]. These measurements are pivotal for the high-precision determinations of both the strong coupling constant $\alpha_s$ [3] and the $W$-boson mass [4, 5], to highlight just two of the most prominent LHC applications within the SM.

In addition to offering important tests of the SM, the $p_{T,Z}$ spectrum can also reveal potential signs of beyond SM (BSM) phenomena. Within the SM, the transverse momentum of the $Z$ boson is mainly generated at lower $p_T$, driven by the recoil of both soft and hard initial-state collinear QCD radiation. However, in the high transverse momentum region, deviations from the SM predictions may arise due to the presence of new particles or interactions not included in the SM. Such scenarios, like the existence of heavy new particles (e.g., extra gauge bosons or dark matter candidates), can result in an excess of DY events with high $p_T$. Under the assumption that these new degrees of freedom cannot be directly produced, deviations of this type can be interpreted in a largely model-independent manner using an effective field theory (EFT) approach, such as the SMEFT [6–9]. Due to their significant phenomenological importance, studies of DY processes have become a central focus of the LHC's SMEFT program — see, for example, the publications [10–22].

Most SMEFT studies on DY production have focused on the impact of four-fermion contact interactions at tree level or loop-level effects from gauge-boson operators, with fewer analyses dedicated to dipole operators [12, 22]. Previous studies have placed bounds on light-quark dipole couplings through electroweak (EW) precision measurements at the $Z$-pole and analyses of EW diboson and DY production at the LHC [12]. Additionally, it has been proposed that this form of BSM physics could help resolve potential discrepancies between theoretical predictions and experimental data for the angular coefficients $A_0$ and $A_2$ [22], whose difference provides a test of the Lam-Tung relation [23–25]. Within the SM, this relation, valid to $\mathcal{O}(\alpha_s)$, implies $A_0 = A_2$. The main aim of this article is to reassess the findings of [12, 22] by utilizing SLC and LEP data on $Z$ production, alongside LHC data on $Z$+jet production. Specifically, we obtain updated constraints on light-quark dipole interactions using precision measurements of the partial $Z$-boson decay widths in $e^+e^-$ collisions and the normalized $p_{T,Z}$ spectrum in NC DY production in $pp$ collisions. With these updated constraints, we evaluate the maximum possible violation of the Lam-Tung relation that such BSM effects could induce. Our results indicate that light-quark dipole operators cannot account for the aforementioned discrepancy.

This work is organised as follows: in Section 2 we detail the theoretical ingredients that are relevant in the context of this article. Our discussion covers the structure of light-quark dipole interactions within the SMEFT framework, an analysis of the BSM modifications of the partial $Z$-boson decay widths and the corresponding $Z$+jet matrix elements, and a concise review of the Lam-Tung relation. The experimental data and the Monte Carlo (MC) setup used to generate the relevant predictions for $Z$+jet production are detailed in Section 3. Our numerical results are presented in Section 4. We derive constraints on light-quark dipole operators using SLC and LEP measurements of EW precision observables, along with current LHC data on NC DY production. The obtained limits are subsequently used to assess the maximum potential impact of this type of SMEFT contributions on the Lam-Tung relation. Section 5 summarizes our key findings and offers

an outlook. This article concludes with a series of appendices. Appendix A contains a comprehensive analysis of the theoretical uncertainties in the $p_{T,Z}$ spectrum in $Z$+jet production within the SM, while Appendix B provides details on the extraction of the Wilson coefficients of the light-quark dipole operators from our fit to the $Z$+jet process.

## 2 Theoretical considerations

In this section, we describe the different theoretical ingredients that are relevant in the context of this article. We start by revisiting the structure of light-quark dipole interactions within the SMEFT framework. Following that, we discuss the modifications of the partial $Z$-boson decay widths, the kinetic properties of the associated $Z$+jet matrix elements, and then review the basics of the Lam-Tung relation.

### Light-quark dipole operators

In the Warsaw operator basis [7], the dimension-six dipole interactions in the SMEFT, involving light-quark fields and EW field strength tensors, are expressed as:

$$
\begin{aligned}
\mathcal{L}_{\text{SMEFT}} \supset{} & \frac{C_{uB}}{\Lambda^2} \bar{Q}_L \sigma^{\mu\nu} u_R \widetilde{H} B_{\mu\nu} + \frac{C_{uW}}{\Lambda^2} \bar{Q}_L \sigma^{\mu\nu} \sigma^a u_R \widetilde{H} W^a_{\mu\nu} \\
& + \frac{C_{dB}}{\Lambda^2} \bar{Q}_L \sigma^{\mu\nu} d_R H B_{\mu\nu} + \frac{C_{dW}}{\Lambda^2} \bar{Q}_L \sigma^{\mu\nu} \sigma^a d_R H W^a_{\mu\nu} + \text{h.c.}
\end{aligned}
\tag{1}
$$

Here, $B_{\mu\nu}$ and $W^a_{\mu\nu}$ denote the $U(1)_Y$ and $SU(2)_L$ field strength tensors, respectively, and $\sigma^a$ are the Pauli matrices. We introduced $\sigma^{\mu\nu} = i/2 \left( \gamma^\mu \gamma^\nu - \gamma^\nu \gamma^\mu \right)$, with $\gamma^\mu$ being the Dirac matrices. The symbol $Q_L = (u_L, d_L)^T$ denotes the left-handed first-generation quark $SU(2)_L$ doublet, while $u_R$ and $d_R$ are the right-handed up-quark and down-quark $SU(2)_L$ singlets. $H$ represents the SM Higgs doublet, and the shorthand notation $\widetilde{H}_i = \epsilon_{ij} \left( H_j \right)^*$, where $\epsilon_{ij}$ is totally antisymmetric with $\epsilon_{12} = 1$, is used. Finally, $\Lambda$ represents the common mass scale that suppresses all operators in (1), making their Wilson coefficients $C_{qB}$ and $C_{qW}$ for $q = u, d$ dimensionless.

After EW symmetry breaking, the light-quark dipole interactions introduced in (1) take the following form

$$
\mathcal{L}_{\text{LEFT}} \supset \frac{v}{\sqrt{2}\Lambda^2} \sum_{q=u,d} \left( C_{q\gamma} \, \bar{q}_L \sigma^{\mu\nu} q_R F_{\mu\nu} + C_{qZ} \, \bar{q}_L \sigma^{\mu\nu} q_R Z_{\mu\nu} \right) + \text{h.c.} \,,
\tag{2}
$$

in the low-energy EFT (LEFT). Here, $v \simeq 246\,\text{GeV}$ is the Higgs vacuum expectation value, while $F_{\mu\nu}$ and $Z_{\mu\nu}$ denote the field strength tensors of the photon and $Z$-boson fields, respectively. The new Wilson coefficients $C_{q\gamma}$ and $C_{qZ}$ are given by the following linear combinations of the original Wilson coefficients $C_{qB}$ and $C_{qW}$:

$$
\begin{aligned}
C_{u\gamma} &= c_w C_{uB} + s_w C_{uW} \,, & C_{d\gamma} &= c_w C_{dB} - s_w C_{dW} \,, \\
C_{uZ} &= -s_w C_{uB} + c_w C_{uW} \,, & C_{dZ} &= -s_w C_{dB} - c_w C_{dW} \,.
\end{aligned}
\tag{3}
$$

The sine and cosine of the weak mixing angle are abbreviated by $s_w \simeq 0.48$ and $c_w \simeq 0.88$, respectively.

Before proceeding, we note that the photonic dipole interactions in (2) are in general more strongly constrained than their $Z$-boson counterparts. The tightest constraints come

from CP-violating observables, such as the neutron electric dipole moment. For instance, the article [26] reports the following upper bounds

$$\frac{|\operatorname{Im} C_{u\gamma}|}{\Lambda^2} < \frac{1}{(64\,\text{TeV})^2}\,, \qquad \frac{|\operatorname{Im} C_{d\gamma}|}{\Lambda^2} < \frac{1}{(185\,\text{TeV})^2}\,, \tag{4}$$

on the magnitudes of the imaginary parts of the Wilson coefficients in (3) involving a photon normalized by two powers of the new-physics scale $\Lambda$. Measurements of the magnetic dipole moments of the neutron and proton [27] place bounds on the real parts of $C_{q\gamma}/\Lambda^2$. However, the resulting limits are both weaker and theoretically less reliable than those provided in (4).

To circumvent the stringent bounds on the Wilson coefficients $C_{q\gamma}$, we consider the following choices of $C_{qB}$ and $C_{qW}$ below:

$$C_u = C_{uB} = -\frac{s_w}{c_w} C_{uW}\,, \qquad C_d = C_{dB} = \frac{s_w}{c_w} C_{dW}\,. \tag{5}$$

For these choices, one has

$$C_{q\gamma} = 0\,, \qquad C_{qZ} = \frac{1}{s_w} C_q\,. \tag{6}$$

and, as a result, the SMEFT modifications of interest in $Z$+jet production can be parameterized by the two coefficients $C_u$ and $C_d$. In the following, we will present all our results in terms of these coefficients. Notice that focusing on the invariant mass region around the $Z$-pole, i.e., $m_{ll} \simeq M_Z$, allows the photon corrections associated with $C_{q\gamma}$ to be neglected in the differential cross section for $Z$+jet production with very high precision. The same holds true for all other observables discussed below. Our BSM predictions are thus largely unaffected by the specific choice of $C_{q\gamma}$ made in (6).

## Partial $Z$-boson decay widths

To derive limits on the light-quark dipole interactions introduced in (1) from the EW precision measurements at SLC and LEP [28], we examine the partial $Z$-boson decay widths. For the specific Wilson coefficients $C_q$ in (5), and treating the light quarks as massless, we find the following modifications

$$\frac{\Gamma\left(Z \to q\bar{q}\right)}{\Gamma\left(Z \to q\bar{q}\right)_{\text{SM}}} = 1 + N_q \frac{v^2 M_Z^2}{\Lambda^4} |C_q|^2\,, \tag{7}$$

where $M_Z \simeq 91.2\,\text{GeV}$ denotes the mass of the $Z$ boson. The factor $N_q$ is flavor dependent and given by

$$N_q = \frac{2}{s_w^2} \frac{1}{g_{Lq}^2 + g_{Rq}^2}\,, \tag{8}$$

with $g_{Lq} = g/c_w \left(T_q^3 - Q_q s_w^2\right)$ and $g_{Rq} = -g/c_w\, Q_q s_w^2$ representing the left-handed and right-handed couplings of the $Z$ boson to quarks of the flavor $q$, respectively — $g$ is the $SU(2)_L$ coupling, whereas $T_q^3$ and $Q_q$ correspond to the third component of the weak isospin and the electromagnetic charge. Explicitly, one has

$$N_u = \frac{18c_w^2}{\pi\alpha\left(9 - 24s_w^2 + 32s_w^4\right)}\,, \qquad N_d = \frac{18c_w^2}{\pi\alpha\left(9 - 12s_w^2 + 8s_w^4\right)}\,. \tag{9}$$

Here $\alpha \simeq 1/128$ denotes the electromagnetic coupling constant. Note that terms linear in $C_q$ are absent in the expression (7) because they would be proportional to the light-quark Yukawa couplings $y_q = \sqrt{2}m_q/v$, which vanish under our assumption. In practice, treating the up and down quarks as massless is an excellent numerical approximation. As a result, the terms quadratic in $|C_q|$ always provide the dominant contribution to the $Z \to q\bar{q}$ decay when considering the effects of light-quark dipole operators.

## Matrix elements for $Z$+jet production

In Section 4, we will derive the constraints that current LHC data on NC DY production impose on the Wilson coefficients of light-quark dipole operators involving a $Z$ boson. While those constraints are derived for the off-shell process $pp \to \gamma^*/Z + X \to l^+l^- + X$, to qualitatively understand those numerical results it is useful to study the analytic structure of the on-shell Born-level matrix elements for the $Z$+jet process. To achieve this, we define the following ratios:

$$\chi_q = \frac{\left|\mathcal{M}_{\mathrm{SM}}(q\bar{q} \to Zg) + \mathcal{M}_{\mathrm{SMEFT}}(q\bar{q} \to Zg)\right|^2}{\left|\mathcal{M}_{\mathrm{SM}}(q\bar{q} \to Zg)\right|^2} . \tag{10}$$

These ratios describe the impact of a non-zero Wilson coefficients $C_q$ in $q\bar{q} \to Zg$ scattering relative to the SM contributions. Relevant diagrams contributing to (10) are shown in Figure 1.

Assuming the light quarks are massless, a straightforward tree-level calculation yields the result

$$\chi_q = 1 + N_q \frac{v^2 M_Z^2}{\Lambda^4} \kappa(s, t) \left|C_q\right|^2 , \tag{11}$$

for the ratios defined in (10). The analytical expressions for the factor $N_q$ are provided in (8) and (9), and $s = \hat{s}/M_Z^2$ and $t = \hat{t}/M_Z^2$ are the usual Mandelstam variables, normalized to the square of the $Z$-boson mass. The kinematic factor appearing in (11) is instead universal and takes the following form:

$$\kappa(s, t) = \frac{s^2 - 4\,(s-1)^2\,t - 4\,(s-1)\,t^2 + 1}{s^2 + 2st + 2\,(t-1)\,t + 1} . \tag{12}$$

A few remarks regarding (10) are in order. First, as in (7), terms linear in $C_q$ are absent in the expression for $\chi_q$, since these interference terms are suppressed by the small Yukawa couplings $y_q$, due to the chirality-flipping nature of the light-quark dipole operators. Second, in the limit of $s \to \infty$, the kinematic factor (12) behaves as

$$\lim_{s \to \infty} \kappa(s, t) = \frac{1 - \cos^2 \hat{\theta}}{1 + \cos^2 \hat{\theta}} \frac{2\hat{s}}{M_Z^2} , \tag{13}$$

where $\hat{\theta}$ represents the scattering angle between the quark and the $Z$ boson in the center-of-mass (CM) frame. The result in (13) demonstrates that the light-quark dipole contributions to DY production are enhanced at high energies compared to the SM background. Similar observations have been made and utilized, for example, in [11, 22, 29–40]. The observed energy enhancement implies that less precise measurements of $p_{T,Z}$ at the LHC can, in principle, achieve similar or even greater sensitivity to $C_q/\Lambda^2$ compared to the high-precision measurements of $Z \to q\bar{q}$ performed at SLC and LEP. Lastly, note that although we have focused on the $q\bar{q} \to Zg$ channel, the same reasoning applies to the processes $qg \to Zq$ and $\bar{q}g \to Z\bar{q}$, which can be derived from $q\bar{q} \to Zg$ through crossing symmetries.

Before proceeding, we finally note that light-quark dipole operators also affect the predictions for differential NC deep inelastic scattering (DIS) measurements, including those conducted at HERA [41]. A simple tree-level calculation shows that the light-quark dipole operators in question give rise to a longitudinal structure function associated with the absorption of a longitudinally polarized virtual $Z$ boson. This results in a violation of the Callan-Gross relation [42] — for an analysis of the effects of quark-lepton contact interactions on the Callan-Gross relation, see [43]. Although these effects are more pronounced for $\hat{t} \to -\infty$, the limited range of four-momentum transfer and experimental uncertainties make current NC DIS measurements inadequate for deriving competitive constraints on the Wilson coefficients of light-quark dipole operators involving a $Z$ boson.

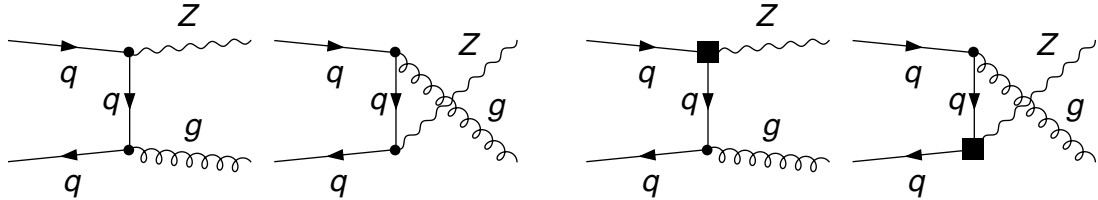

Figure 1: Contributions to the $q\bar{q} \to Zg$ process in the SM (left) and the SMEFT (right). The black squares represent an insertion of a light-quark dipole operator involving a $Z$ boson, as described by the Lagrangian (2).

## Lam-Tung relation and its violation

The Lam-Tung relation is a theoretical prediction related to the angular distribution of dileptons produced in the NC DY process, i.e., $pp \to \gamma^*/Z + X \to l^+l^- + X$. In the so-called Collins-Soper (CS) frame [44], the angular distribution of the outgoing leptons can be described in terms of a set of angular coefficients $A_i$ for $i = 0, \ldots, 7$. These frame-dependent angular coefficients provide a detailed description of the lepton angular distributions as functions of the momenta of the exchanged gauge boson. Using these angular coefficients, the differential cross section can be expressed as:

$$\frac{d\sigma}{dp_{T,ll}\,dy_{ll}\,dm_{ll}^2\,d\Omega} = \frac{3}{16\pi}\frac{d\sigma}{dp_{T,ll}\,dy_{ll}\,dm_{ll}^2}\left[\left(1+\cos^2\theta\right) + \frac{A_0}{2}\left(1 - 3\cos^2\theta\right)\right.$$

$$\left. + A_1\,\sin 2\theta \cos\phi + \frac{A_2}{2}\sin^2\theta\cos 2\phi + A_3\,\sin\theta\cos\phi \right. \tag{14}$$

$$\left. + A_4\,\cos\theta + A_5\,\sin^2\theta\sin 2\phi + A_6\,\sin 2\theta\sin\phi + A_7\,\sin\theta\sin\phi\right].$$

Here, $d\Omega = d\cos\theta\,d\phi$ with $\theta$ and $\phi$ denoting the polar and azimuthal angles of the negatively charged lepton in the CS frame, while $p_{T,ll}$, $y_{ll}$, and $m_{ll}$ represent the transverse momentum, rapidity, and invariant mass of the lepton pair, respectively. It should be noted that $p_{T,ll}$ is a good proxy for $p_{T,Z}$ when events are limited to the dilepton invariant mass region near $m_{ll} \simeq M_Z$, as is typically done in experimental studies. In terms of the angular coefficients $A_i$ introduced in (14), the Lam-Tung relation reads:

$$A_0 - A_2 = 0. \tag{15}$$

This relation holds up to $\mathcal{O}(\alpha_s)$ in perturbative QCD under the leading-twist approximation. It arises from the fact that the NC DY process involves at leading-order (LO) in QCD the annihilation of spin-1/2 quarks and antiquarks, and is further preserved at next-to-leading order (NLO) in QCD due to the purely vectorially coupling of the spin-1 gluon to the quark current [23–25,45]. In the SM, the equality (15) is first violated in perturbation theory by $\mathcal{O}(\alpha_s^2)$ corrections [46], though the breaking of the Lam-Tung relation remains relatively small. The relation (15) therefore provides a clear prediction for the behavior of the angular coefficients $A_0$ and $A_2$ in the NC DY process, and its experimental study can help in understanding the intricate dynamics that triggers the $pp \to \gamma^*/Z + X \to l^+l^- + X$ process both in the SM and beyond it.

The effects of various dimension-six and dimension-eight SMEFT contributions on the Lam-Tung relation have been explored in [10,13,22]. To gain insight on how the light-quark dipole operators involving a $Z$ boson modify the predictions for the angular distributions,

we integrate the angular distribution (14) over the azimuthal angle $\phi \in [0, 2\pi]$. This yields

$$\frac{d\sigma}{dp_{T,ll}\,dy_{ll}\,dm_{ll}^2\,d\cos\theta} = \frac{3}{8}\frac{d\sigma}{dp_{T,ll}\,dy_{ll}\,dm_{ll}^2}\left[\left(1 + \frac{A_0}{2}\right)\left(1 + a_0\cos^2\theta\right) + A_4\cos\theta\right]. \quad (16)$$

Integrating (14) over the polar angle $\cos\theta \in [-1, 1]$ instead results in:

$$\frac{d\sigma}{dp_{T,ll}\,dy_{ll}\,dm_{ll}^2\,d\phi} = \frac{1}{2\pi}\frac{d\sigma}{dp_{T,ll}\,dy_{ll}\,dm_{ll}^2}\left[1 + a_2\cos 2\phi + \frac{3\pi}{16}A_3\cos\phi \right.$$
$$\left. + \frac{A_5}{2}\sin 2\phi + \frac{3\pi}{16}A_7\sin\phi\right]. \quad (17)$$

Here, we have introduced the abbreviations

$$a_0 = \frac{2 - 3A_0}{2 + A_0}, \qquad a_2 = \frac{A_2}{4}. \quad (18)$$

Note that $A_0 = 0$ necessarily implies $a_0 = 1$, and the reverse is also true.

The formulas (16) and (17) can provide some insight on how BSM contributions can modify the expected angular dependence. Let us start by examining the case of the tree-level photon contribution to NC DY production at the LHC for a single lepton flavor, i.e., the $q\bar{q} \to \gamma^* \to l^+l^-$ process. In the limit of massless external fermions, the corresponding differential cross section is given by

$$\frac{d\sigma}{d\hat{\Omega}} = \frac{\alpha^2 Q_q^2}{12\hat{s}}\left(1 + \cos^2\hat{\theta}\right), \quad (19)$$

where $Q_u = 2/3$ and $Q_d = -1/3$ represent the electromagnetic charge factors of the up and down quark, respectively. Recalling that for a $2 \to 2$ process the CM and the CS frame are identical, a comparison of (19) with (16) gives

$$A_0 = 0, \qquad A_2 = 0. \quad (20)$$

The same reasoning can also be applied to the light-quark dipole contributions to $q\bar{q} \to Z \to l^+l^-$ scattering. In this case, we find

$$\frac{d\sigma}{d\hat{\Omega}} = \frac{\alpha\left(1 - 4s_w^2 + 8s_w^4\right)v^2\,|C_q|^2}{192\pi\,s_w^4 c_w^2\Lambda^4}\frac{\hat{s}^2}{\left(\hat{s} - M_Z^2\right)^2}\left(1 - \cos^2\hat{\theta}\right), \quad (21)$$

when all external fermions are treated as massless, implying

$$A_0 \neq 0, \qquad A_2 = 0. \quad (22)$$

This shows that at the tree level, the light-quark dipole operators lead to changes in the prediction for the angular coefficient $A_0$, which already imply $A_0 \neq A_2$. It is important to note that for $p_{T,ll} = 0$, the azimuthal symmetry of the scattering process dictates that $A_2 = 0$ [44]. Therefore, in both the SM and BSM scenarios, the condition $A_2 = 0$ must hold for these simple cases. Observe that the angular dependence in (19) and (21) resembles that of the denominator and numerator of (13), respectively. We add that the polar angle dependence found in (19) and (21) can also be understood by analyzing the chiralities or helicities of the external fermions involved in the underlying scattering processes. In fact, the sign difference in the $\cos^2\hat{\theta}$ term is due to the dipole interactions flipping the chiralities of the incoming quarks, whereas the photon couples exclusively to quarks and antiquarks with the same chirality.

# 3  Data and MC predictions for $Z$+jet production

To derive constraints on $C_q/\Lambda^2$, we utilize the measurement of the normalized $p_{T,ll}$ distribution of DY lepton pairs reported by ATLAS [1]. These measurements were performed using LHC data collected at a CM energy of $\sqrt{s} = 13\,\text{TeV}$, with an integrated luminosity of $36.1\,\text{fb}^{-1}$. Both dimuon and dielectron final states are analyzed within a dilepton invariant mass window of $66\,\text{GeV} < m_{ll} < 116\,\text{GeV}$. The results are presented within a fiducial phase space designed to be close to the experimental acceptance, defined by lepton transverse momenta $p_{T,l} > 27\,\text{GeV}$ and lepton pseudorapidities $|\eta_l| < 2.5$.

A key ingredient for the extraction of the constraints on $C_q/\Lambda^2$ is the SM prediction for the normalized $p_{T,ll}$ spectrum, which serves as a background for the BSM contributions. For this, we utilize the SM prediction published alongside the experimental results in [1]. This SM prediction is based on an NNLO QCD prediction for the $Z$+jet process at $\mathcal{O}(\alpha_s^3)$ obtained with NNLOJET [47, 48], further supplemented by NLO EW effects [49]. The reported SM prediction incorporates an estimate of the theoretical uncertainty based on the complete NNLO QCD calculation. As outlined in Appendix A, we independently evaluated the potential effects of uncertainties arising from scale variations,[1] the choice of $\alpha_s(M_Z)$, and the input parton distribution functions (PDFs). These uncertainties were subsequently combined to determine the total theoretical uncertainties, which were found to be slightly larger than those reported by ATLAS. Consequently, the limits we derive on $C_q/\Lambda^2$ are more conservative than those that would be obtained by relying solely on the ATLAS evaluation of the normalized $p_{T,ll}$ spectrum and their associated uncertainty estimates. The light-quark dipole corrections to the normalized $p_{T,ll}$ spectrum in $Z$+jet production are calculated by means of a FeynRules 2.0 [50] implementation of the Lagrangian (1) in the UFO format [51]. The corresponding $Z$+jet events were generated at LO in QCD using MadGraph5_aMCNLO [52], followed by parton showering with PYTHIA 8.2 [53]. The protons in the LHC collisions were modeled using the NNPDF31_nnlo_as_01180 [54] PDFs. To efficiently generate SMEFT events with high $p_{T,ll}$, we applied a generation bias in the form $(p_{T,j}/\text{TeV})^2$, where $p_{T,j}$ is the $p_T$ of the jet recoiling against the $Z$ boson. This approach was employed to enhance the statistical precision of our BSM samples. The BSM samples were subjected to the experimental cuts outlined at the beginning of this section. To ensure that our MC generation does not introduce bias into our results, we computed the high-$p_{T,ll}$ tail of the $p_{T,ll}$ spectrum for $Z$+jet production at LO QCD within the SM using our setup. The results show agreement at the percent level with the $\mathcal{O}(\alpha_s)$ prediction obtained using NNLOJET. Note that, since the interference term between the SM and BSM contributions vanishes in the limit of massless external fermions, the two predictions for the normalized $p_{T,ll}$ distributions can be directly added to obtain the full SM plus BSM results, with the BSM contributions calculated at $\mathcal{O}(\alpha_s)$.

As previously discussed, we will then use the derived constraints on $C_q/\Lambda^2$ to evaluate the potential impact that these BSM contributions may have on the prediction of the angular coefficients in the NC DY process. These angular coefficients have been measured in the vicinity of the $Z$-boson mass peak in [55]. The data analyzed correspond to $20.3\,\text{fb}^{-1}$ of LHC collisions at a CM energy of $\sqrt{s} = 8\,\text{TeV}$. In that analysis, the lepton pair is required to have an invariant mass within the window of $80\,\text{GeV} < m_{ll} < 100\,\text{GeV}$, while results are made available differential in $y_{ll}$ and $p_{T,ll}$ (up to $600\,\text{GeV}$). The analysis [55] measures all eight angular coefficients $A_i$ and finds deviations between the SM prediction (calculated at $\mathcal{O}(\alpha_s^2)$) and the measured values for the difference $A_0 - A_2$ in the tail of the $p_{T,ll}$ spectrum, suggesting that either higher-order QCD or EW corrections, or BSM

---

[1]We thank Alexander Huss for providing us with the SM predictions up to NNLO in QCD, obtained using NNLOJET, as well as the breakdown of factorization and renormalization scale variations.

physics, are necessary to accurately describe the data. Although less significant, a similar trend was also observed by CMS in [56]. A phenomenological analysis of these angular coefficients was subsequently performed in [57], including the impact of QCD corrections up to $\mathcal{O}(\alpha_s^3)$, which relied on the calculations presented in [47, 48]. Those corrections reduced the quoted tension but still fail to accurately describe the data for the difference $A_0 - A_2$ at high $p_{T,ll}$.

To evaluate the impact of light-quark dipole operators on the angular coefficients $A_i$, we calculated the relevant $q\bar{q} \to l^+l^- + g$ tree-level matrix elements, as well as the crossed channels, using a combination of the `FeynArts` [59] and `FormCalc` [60] packages. The resulting analytic expressions were subsequently incorporated into a private code for NC DY production, developed in the context of [61]. The BSM predictions were produced using `PDF4LHC15_nnlo_30` PDFs, the same choice of EW inputs as in [57], and were required to satisfy the same analysis criteria imposed by ATLAS in [55]. To determine the angular coefficients $A_i$, we employ projectors (see, for example, [57]). This approach involves calculating normalized weighted averages over the angular variables $\theta$ and $\phi$. The normalization is based on the LO QCD prediction of the $p_{T,ll}$ spectrum in $Z$+jet production. Finally, note that the SM and BSM contributions to the angular coefficients $A_i$ are additive, as the corresponding amplitudes do not interfere when the external fermions are taken to be massless. This allows the LO BSM predictions for the difference of the angular coefficients $A_0$ and $A_2$ to be directly added to the SM prediction, which is available at $\mathcal{O}(\alpha_s^3)$ from [57].

# 4 Numerical results

In this section, we derive the constraints on the Wilson coefficients of the light-quark dipole operators using the partial $Z$-boson decay widths and the normalized $p_{T,ll}$ spectrum in $Z$+jet production. With the derived limits, we then evaluate the maximum possible impact that this type of BSM effects could have on the angular coefficients relevant to the Lam-Tung relation.

## EW precision measurements

The light-quark dipole interactions (1) can be constrained through the EW precision measurements conducted at the $Z$-pole. In the following, we analyze the partial $Z$-boson decay widths. Without assuming lepton universality, the partial decay width of the $Z$ boson into light quarks has been measured with a precision of 5.9‰ at the 95% confidence level (CL) [27, 28]. This measurement, combined with the results in (7), imposes the following condition:

$$\frac{(0.46\,\text{TeV})^2}{\Lambda^2}\sqrt{|C_u|^2 + 0.93\,|C_d|^2} < 7.6 \cdot 10^{-2}\,. \tag{23}$$

Assuming only one Wilson coefficient is non-zero at a time, this inequality leads to the constraints:

$$\frac{|C_u|}{\Lambda^2} < \frac{1}{(1.7\,\text{TeV})^2}\,, \qquad \frac{|C_d|}{\Lambda^2} < \frac{1}{(1.6\,\text{TeV})^2}\,. \tag{24}$$

## Normalized $p_{T,ll}$ distribution

In Figure 2, we compare the results of ATLAS [1] with different predictions for the normalized $p_{T,ll}$ spectrum in $Z$+jet production. The uncertainties of the measurement are

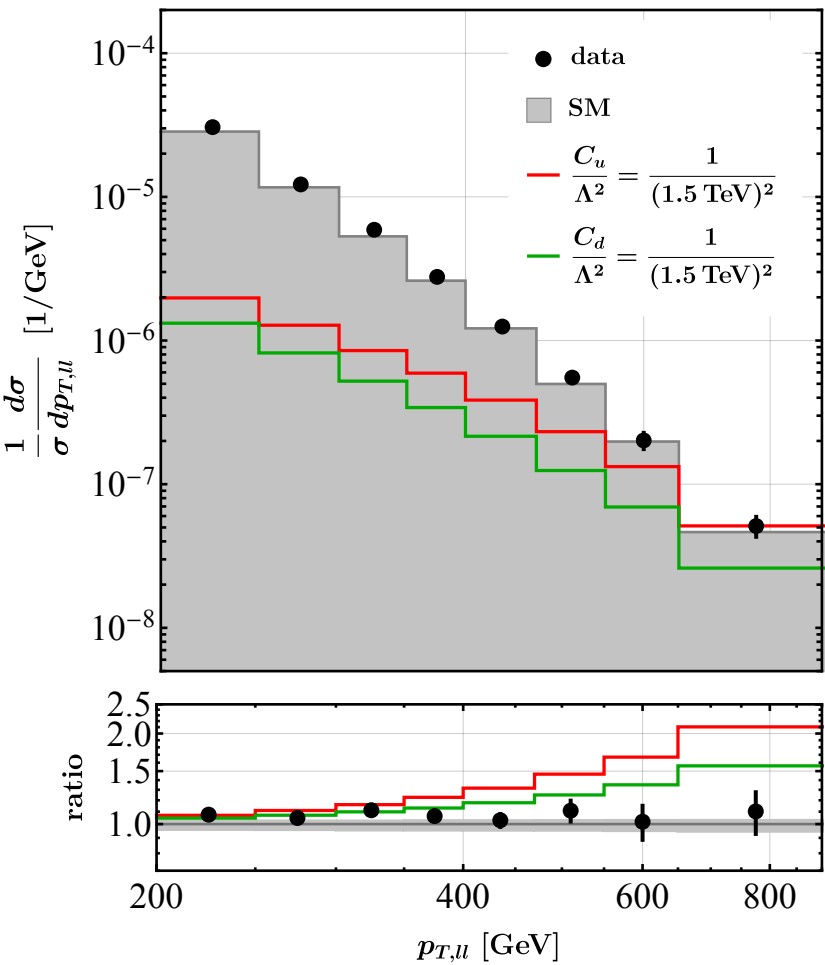

Figure 2: Comparison of the normalized $p_{T,ll}$ distribution for $p_{T,ll} \in [200, 900]\,\text{GeV}$. Black points represent the ATLAS measurement [1], with statistical uncertainties depicted as black bars. The gray histogram corresponds to the SM prediction, and its systematic uncertainties are shown in the lower ratio plot as a gray band. Predictions for the BSM effects are displayed for $C_u/\Lambda^2 = 1/(1.5\,\text{TeV})^2$ and $C_d/\Lambda^2 = 1/(1.5\,\text{TeV})^2$, and depicted as red and green curves, respectively. The other Wilson coefficient not indicated is set to zero. Further details are provided in the main text.

represented by black bars. The SM prediction is represented by a gray histogram, with its uncertainties illustrated as a gray band in the lower ratio plot. These predictions are taken from [1], where they were calculated using NNLOJET [47, 48] and supplemented with NLO EW corrections [49]. The red and green curves represent our BSM predictions for the normalized $p_{T,ll}$ distribution for the two choices $C_u/\Lambda^2 = 1/(1.5\,\text{TeV})^2$ and $C_d/\Lambda^2 = 1/(1.5\,\text{TeV})^2$ of Wilson coefficients introduced in (5). These results were generated using the MC setup detailed in Section 3. In line with (13), we observe that the considered BSM effects become more pronounced relative to the SM background at high $p_{T,ll}$. For example, in the highest bin, $p_{T,ll} \in [650, 900]\,\text{GeV}$, the benchmark values of $C_q/\Lambda^2$ result in enhancements of approximately 210% and 160% relative to the SM prediction. It is evident from the figure that such significant enhancements are ruled out by the ATLAS data.

We proceed to derive 95% CL limits on $C_q/\Lambda^2$, using the results for the normalized $p_{T,ll}$ spectrum in $Z$+jet production presented in Figure 2. The significance is determined as a

ratio of Poisson likelihoods, adjusted to account for the statistical and systematic uncertainties on the background reported in [1] and a systematic uncertainty of 5% on the BSM predictions. In Appendix A, we assess the systematic uncertainty of the SM prediction. Our evaluation yields slightly larger uncertainties than those reported in [1]. As a result, the limits we derive on the combinations $C_q/\Lambda^2$ are more conservative than those that would be obtained using only the ATLAS evaluation of the normalized $p_{T,ll}$ spectrum and its associated uncertainty estimates. All systematic uncertainties are incorporated as Gaussian constraints [62]. Our likelihood analysis leads to the following inequality:

$$\frac{(0.46\,\text{TeV})^2}{\Lambda^2}\sqrt{|C_u|^2 + 0.51\,|C_d|^2} < 4.1 \cdot 10^{-2}\,. \tag{25}$$

Observe that we have expressed (25) in a form similar to (23) to illustrate that current LHC measurements of the high-$p_{T,ll}$ spectrum in DY production exhibit a sensitivity to $C_q/\Lambda^2$ that is already stronger by a factor of approximately 1.8 (1.4) compared to the SLC and LEP measurements of the partial decay width of the $Z$ boson into up quarks (down quarks). Under the assumption that only one Wilson coefficient is non-zero, the condition (25) translates into the following upper limits:

$$\frac{|C_u|}{\Lambda^2} < \frac{1}{(2.3\,\text{TeV})^2}\,, \qquad \frac{|C_d|}{\Lambda^2} < \frac{1}{(1.9\,\text{TeV})^2}\,. \tag{26}$$

A few comments appear to be appropriate. First, since the SM predictions in Figure 2 consistently fall below the data, the limits (26) depend on how many $p_{T,ll}$ bins are included in the likelihood analysis. The values reported above are based on the two highest $p_{T,ll}$ bins from the ATLAS measurements [1], as this choice (see Appendix B) yields the most stringent upper limits on $|C_q|/\Lambda^2$. We observe that our likelihood analysis also results in lower bounds with $|C_q|/\Lambda^2 > 0$, which challenge the SM. However, these bounds arise from the SM prediction for the normalized $p_{T,ll}$ spectrum consistently falling below the ATLAS data by an almost constant offset, whereas the considered BSM effects would scale as $p_{T,ll}^2$. Thus, we restricted our analysis to determining only upper limits on $|C_q|/\Lambda^2$, following [62], since these are less affected by the mismatch between the SM prediction and the data than the lower bounds.

Second, the limits in (26) are stronger than those in (24), a result consistent with the findings of [12]. The bounds in (26) are primarily constrained by the limited statistics in the current ATLAS data on $Z$+jet production at high $p_{T,ll}$. Since this measurement uses only $36.1\,\text{fb}^{-1}$ of data, naive luminosity scaling suggests that at the high-luminosity option of the LHC (HL-LHC), with $3000\,\text{fb}^{-1}$ of integrated luminosity, an improvement by a factor of approximately 3 can be expected concerning (26). To make this statement more precise, we repeated our likelihood analysis using simulated SM and BSM samples, assuming a CM energy of $\sqrt{s} = 14\,\text{TeV}$ and considering $p_{T,ll}$ values in the range $[600, 3000]\,\text{GeV}$. We assumed a 5% systematic uncertainty for both the SM and the BSM predictions. Our analysis indicates that the bounds in (26) could potentially be improved by a factor of about 4.5 at the HL-LHC. The improvement exceeds naive expectations due to the quadratic energy growth of the light-quark dipole corrections to the $p_{T,ll}$ distribution (see (13)), which offsets the suppression from the decreasing parton luminosities.

## Violation of Lam-Tung relation

Using the limits in (24) and (26), we can now evaluate the maximum effect of light-quark dipole operators on the prediction of the difference between the angular coefficients $A_0$ and $A_2$, and thus assess their potential to explain the observed violation of the Lam-Tung

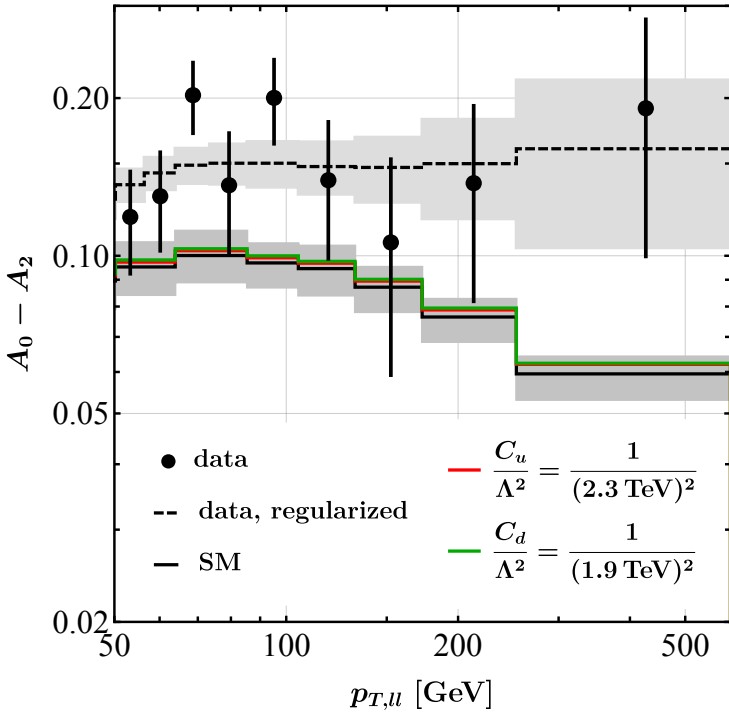

Figure 3: Comparison of the predictions for $A_0 - A_2$ in the range $p_{T,ll} \in [50, 600]$ GeV. The black points show the central values of the ATLAS measurement [55], with statistical uncertainties represented by black error bars. The black solid curve and gray band represent the SM prediction and its systematic uncertainties. Predictions including BSM effects are shown for $C_u/\Lambda^2 = 1/(2.3\,\text{TeV})^2$ and $C_d/\Lambda^2 = 1/(1.9\,\text{TeV})^2$, depicted as red and green curves, respectively. The other Wilson coefficient not specified in the figure is set to zero. The regularized ATLAS data is also included and shown as a dashed black line and a light gray band. Additional details are given in the main text.

relation, which exceeds the NNLO QCD prediction within the SM. In Figure 3, we show our results for the combination $A_0 - A_2$ of angular coefficients as a function of $p_{T,ll}$ for $pp$ collisions at a CM energy of $\sqrt{s} = 8$ TeV. The ATLAS data [55] is shown as black points with error bars. The SM prediction is depicted by a solid black line, with its uncertainties represented by a gray band. As detailed in Section 3, these predictions are based on the NNLO QCD calculation performed in [57]. The plot reveals a tendency for the ATLAS data to systematically exceed the SM prediction for $A_0 - A_2$ at higher $p_{T,ll}$ values. A $\chi^2$ test performed in [57], however, demonstrated that the SM prediction aligns reasonably well with the ATLAS measurement across all 38 data points, yielding $\chi^2/38 = 1.8$. For a better comparison with the recent article [22], we have included in Figure 3 also the regularized ATLAS data, shown as a dashed black line with gray shading — details on the experimental regularization procedure can be found in Appendix C of [55]. For the purposes of this work, it is sufficient to note that the regularization procedure introduces large bin-to-bin correlations in the distributions of the angular coefficients $A_i$. A visual comparison with the regularized data can therefore be misleading, as these strong correlations are not apparent in the regularized histograms. For $A_0 - A_2$, the figure clearly illustrates that a naive comparison of the SM prediction with the regularized ATLAS data will lead to an overestimation of the disagreement between theory and experiment.

The red and green curves in Figure 3 illustrate our $A_0 - A_2$ predictions for the two choices of Wilson coefficients, $C_u/\Lambda^2 = 1/(2.3\,\text{TeV})^2$ and $C_d/\Lambda^2 = 1/(1.9\,\text{TeV})^2$, cor-

responding to the upper limits given in (26). Note that the value of the Wilson coefficient for the up-quark dipole operator considered in the paper [22] corresponds to $C_u/\Lambda^2 = s_w/\text{TeV}^2 \simeq 1/(1.4\,\text{TeV})^2$. Since the BSM contributions to $A_0 - A_2$ scale as $(|C_q|/\Lambda^2)^2$, the choice of Wilson coefficient for the up-quark dipole operator in [22] results in relative effects approximately six times larger than those displayed in Figure 3. The figure shows that the inclusion of the considered light-quark dipole contributions has only a very minor impact on the $A_0 - A_2$ distribution, staying within the uncertainty band of the SM prediction. Consequently, the resulting BSM effects are clearly insufficient to bridge the gap between theory and experiment in the final bin, spanning the range $[253, 600]\,\text{GeV}$, of the ATLAS measurement. The $C_q/\Lambda^2$ values shown in the plot therefore nicely highlight that, given current experimental constraints, BSM effects as described by (1) are essentially ruled out as a solution to the tension between the SM prediction and the ATLAS measurement at large $p_{T,ll}$.

## 5 Conclusions

In this article, we conducted model-independent analyses of potential BSM modifications in NC DY production in both $e^+e^-$ and $pp$ collisions, focusing on light-quark dipole operators arising at the dimension-six level within the SMEFT framework. These analyses allowed us to revisit the findings of [12, 22]. Previous studies have derived constraints on light-quark dipole couplings using EW precision measurements at the $Z$-pole and analyses of EW diboson and DY production at the LHC [12]. Additionally, this type of BSM physics has been proposed as a possible explanation for the observed discrepancies between theoretical predictions and experimental measurements of the violation of the Lam-Tung relation in the tail of the $p_{T,ll}$ spectrum in $Z$+jet production [22].

We began our numerical analysis by deriving constraints on the Wilson coefficients of the light-quark dipole operators using precision measurements of the $Z$-boson decay width performed at SLC and LEP, as well as the normalized $p_{T,ll}$ distribution measured in $pp \to \gamma^*/Z + X \to l^+l^- + X$ production at the LHC. The latter bounds were computed using existing state-of-the-art predictions for NC DY production within the SM, incorporating NNLO QCD corrections [47, 48] and NLO EW effects [49], alongside a conservative estimate of theoretical uncertainties — details of which can be found in Appendix A. We observed that the effects of light-quark dipole operators are enhanced at high energies, and as a result, the precision of current $p_{T,ll}$ measurements at the LHC already exceeds the sensitivity of the high-precision $e^+e^- \to Z \to q\bar{q}$ measurements at SLC and LEP. This conclusion is in agreement with [12].

Next, we applied the obtained constraints to evaluate the maximum impact that light-quark dipole operators could have on the predictions for the angular coefficients $A_0$ and $A_2$, which appear in the Lam-Tung relation (15). Our BSM predictions for the difference $A_0 - A_2$ were derived at LO and combined with the existing SM predictions for the angular coefficients $A_i$ in $Z$-boson production at NNLO in QCD [47, 48, 57], which represent the most advanced SM calculations currently available. We found that the limits (24) and (26) on the Wilson coefficients of the light-quark dipole operators exclude the parameter space that could account for the discrepancy between theoretical predictions and experimental observations of the Lam-Tung relation, as reported by the ATLAS collaboration in [55]. This finding contrasts with the results of the recent work [22], which relied on values for the Wilson coefficients that are inconsistent with both (24) and (26). Therefore, BSM effects of the type (1) are essentially ruled out as the cause of the excess observed in the difference $A_0 - A_2$ at high $p_{T,ll}$. This suggests that other factors, such as unaccounted-for

QCD, EW, or experimental effects, are more likely explanations at present.

We also noted that the constraints in (26) on the Wilson coefficients of the light-quark dipole operators, derived from current LHC data on the normalized $p_{T,ll}$ spectrum in $Z$+jet production, are limited by statistics. This is because the bounds are primarily driven by the high $p_{T,ll}$ bins, where the BSM contributions have a larger impact compared to the SM. As a result, future measurements of NC DY production during the HL-LHC could potentially improve the constraints presented in (26) by a factor of about 4.5. Similar statements apply to the HL-LHC measurements of all angular coefficients $A_i$, which are currently not well measured in the high-$p_{T,ll}$ regime due to insufficient statistics. Precise future experimental measurements of the Lam-Tung relation could therefore offer additional and partially complementary insights into BSM physics, especially if deviations are detected at high $p_{T,ll}$ in the unpolarized differential cross section.

# Acknowledgements

We thank Alexander Huss for providing us with both the SM predictions for the differential $p_{T,ll}$ spectrum and the fiducial cross section in NC DY production, obtained using `NNLOJET`, including the breakdown of factorization and renormalization scale variations. We also acknowledge Luc Schnell's support in event generation with `MadGraph5_aMCNLO`. The Feynman diagrams shown in this work were generated and drawn with `FeynArts`. JW is part of the International Max Planck Research School (IMPRS) on "Elementary Particle Physics".

# A   SM prediction for $Z$+jet production

In this appendix, we estimate the theoretical uncertainties affecting the $p_{T,ll}$ spectrum in $Z$+jet production within the SM. These uncertainties arise from various sources, including variations in the renormalization and factorization scales, uncertainties in the value of the strong coupling constant $\alpha_s(M_Z)$ at the scale $M_Z$, and uncertainties associated with the chosen PDF set. Each of these sources of theoretical uncertainty will be discussed individually before determining the total theoretical uncertainties in the $p_{T,ll}$ distribution.

### Scale uncertainties

The scale uncertainties of the $p_{T,ll}$ spectrum in the SM are evaluated using the $Z$+jet production results at $\mathcal{O}(\alpha_s^3)$, as implemented in `NNLOJET` [47,48]. One of the authors of these studies provided the `NNLOJET` predictions for the $p_{T,ll}$ spectrum and fiducial cross section, including variations in factorization and renormalization scales. Similar results were also shared with ATLAS for the measurement reported in [1]. Since these results play a crucial role in determining the limits presented in Section 4, we summarize their key details below. The predictions were derived using the central member of the `NNPDF31_nnlo_as_01180` PDF set. In this calculation, the renormalization and factorization scales were dynamically set on an event-by-event basis to

$$\mu_R = k\,E_{T,Z}\,, \qquad \mu_F = m\,E_{T,Z}\,, \tag{A.1}$$

where $E_{T,Z} = \sqrt{m_{ll}^2 + p_{T,ll}^2}$ represents the transverse energy of the virtual $Z$ boson. Seven combinations of $k$ and $m$ are considered:

$$\{k,m\} \in \left\{\{1,1\},\{0.5,0.5\},\{2,2\},\{1,0.5\},\{1,2\},\{0.5,1\},\{2,1\}\right\}. \tag{A.2}$$

The scale choice $\{k, m\} = \{1, 1\}$ is used to define the central values of the predictions, including the total cross section and the $p_{T,ll}$ distribution. The theoretical uncertainties due to scale variations are determined by taking the envelope of the predictions corresponding to all seven combinations in (A.2). For instance, in the case of the total $Z$-boson production cross section at $\mathcal{O}(\alpha_s^2)$ at a CM energy of $\sqrt{s} = 13\,\mathrm{TeV}$, within the fiducial region defined at the beginning of Section 3, the following result for a single lepton flavor is obtained:

$$\sigma = 728.7 \left(1^{+0.42\%}_{-0.72\%}\right) \mathrm{pb}\,. \tag{A.3}$$

The same approach, when applied to the case of the $p_{T,ll}$ spectrum, yields relative variations of about $^{+2\%}_{-3\%}$ within the $p_{T,ll}$ range of interest. This feature is illustrated in Figure 4 by the solid red lines.

### Uncertainties related to $\alpha_s$

The uncertainties associated with the choice of the strong coupling constant $\alpha_s$ defined at the scale $M_Z$ are estimated by generating $Z$+jet event samples at LO in QCD using `MadGraph5_aMCNLO` with various PDF fits from `NNPDF31_nnlo`, each obtained with a different value of $\alpha_s(M_Z)$. The following eleven choices are considered:

$$\alpha_s(M_Z) \in \big\{0.108, 0.110, 0.112, 0.114, 0.116, 0.117, 0.118, 0.119, 0.120, 0.122, 0.124\big\}\,. \tag{A.4}$$

The uncertainties of the predictions are derived by applying a polynomial fit to the results, considering the variations in the strong coupling constant $\alpha_s(M_Z) = 0.1180 \pm 0.0009$ [27]. The central values of the predictions are identified with the values obtained for the central member of the `NNPDF31_nnlo_as_01180` PDF set. Using the fiducial cross section of NC DY production at a CM energy of $\sqrt{s} = 13\,\mathrm{TeV}$ as an example, we find that this method yields the following prediction for a single lepton flavor:

$$\sigma = 710.1 \left(1^{+0.37\%}_{-0.65\%}\right) \mathrm{pb}\,. \tag{A.5}$$

As depicted by the dotted green curves in Figure 4, this approach results in relative uncertainties of less than about $\pm 0.5\%$ for the $p_{T,ll}$ distribution, which are associated with the choice of the strong coupling constant $\alpha_s$.

### PDF uncertainties

The PDF uncertainties are determined by generating $Z$+jet predictions at LO in QCD using `MadGraph5_aMCNLO` for all 100 members of the `NNPDF31_nnlo_as_01180` PDF set. Assuming a Gaussian distribution, the central values of our predictions are determined from the results corresponding to the central member of the `NNPDF31_nnlo_as_01180` PDF set, while the uncertainties are calculated from the standard deviations relative to the average of all 100 members. Applying this methodology to the fiducial cross section for NC DY production at a CM energy of $\sqrt{s} = 13\,\mathrm{TeV}$ gives

$$\sigma = 710.1 \left(1^{+0.85\%}_{-0.74\%}\right) \mathrm{pb}\,, \tag{A.6}$$

for a single lepton flavor. In the case of the $p_{T,ll}$ spectrum, the corresponding PDF uncertainties are depicted as dashed blue lines in Figure 4. We find that the relative uncertainties due to the choice of PDFs are approximately $\pm 1\%$ at low $p_{T,ll}$ values, rising to around $\pm 1.5\%$ in the high-$p_{T,ll}$ tail of the shown $Z$+jet results.

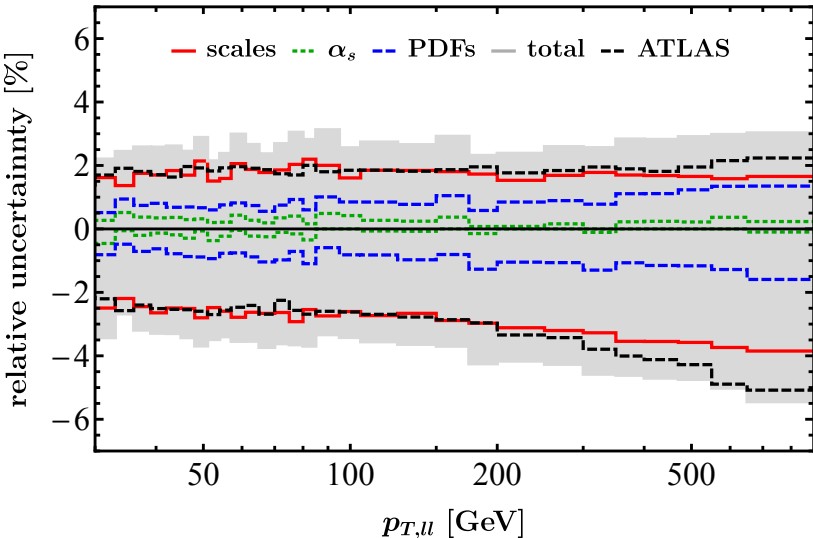

Figure 4: The individual uncertainties and their combined effect for the $p_{T,ll}$ spectrum in $Z$+jet production at the LHC with a CM energy of $\sqrt{s} = 13\,\text{TeV}$. The uncertainties due to scale variations, $\alpha_s$, and PDFs are illustrated by the solid red, dotted green, and dashed blue lines, respectively, while the total combined uncertainties are represented by the gray band. For comparison, the total uncertainties reported by the ATLAS measurement [1] are shown as dashed black lines.

### Combined theoretical uncertainties

We calculate the total theoretical uncertainties by combining the uncertainties from $\alpha_s$ and the PDFs in quadrature, while adding the uncertainties from scale variations linearly. Note that combining the uncertainties from $\alpha_s$ and the PDFs in quadrature aligns with the recommendations for using PDF sets as formulated in [58]. In fact, assuming Gaussian distributions and linear error propagation, adding the $\alpha_s$ and PDF uncertainties in quadrature automatically accounts for the correlation between these two sources of uncertainties [63], making this approach well justified. In contrast, adding the scale uncertainties linearly to the combined $\alpha_s$ and PDF uncertainties is simply one option. This approach is justified by recognizing that scale uncertainties are systematic and non-Gaussian in nature.

Figure 4 presents the individual uncertainties and their combination for the $p_{T,ll}$ spectrum in $Z$+jet production, assuming LHC collisions at a CM energy of $\sqrt{s} = 13\,\text{TeV}$. We observe that scale variations dominate the uncertainties across all bins, with PDF uncertainties becoming slightly more important at higher $p_{T,ll}$ values. As a result, the total theoretical uncertainties grow from $^{+2.2\%}_{-3.4\%}$ at $p_{T,ll} = 30\,\text{GeV}$ to $^{+3.0\%}_{-5.4\%}$ at $p_{T,ll} = 600\,\text{GeV}$. The ATLAS analysis [1] instead quotes total theoretical uncertainties of $^{+1.7\%}_{-2.2\%}$ and $^{+2.2\%}_{-5.1\%}$ for the same $p_{T,ll}$ values — the total uncertainties from the ATLAS measurement [1] are displayed as dashed black lines in Figure 4. These numbers imply that, when symmetrized, our uncertainty estimates are approximately 20% larger than those reported by ATLAS. The constraints we established in Section 4 on $|C_q|/\Lambda^2$ therefore turn out to be more conservative than those that would derive from the ATLAS evaluation of the theoretical uncertainties affecting the normalized $p_{T,ll}$ spectrum. We note that the theoretical uncertainties associated with the EW corrections to $Z$+jet production are not accounted for in our uncertainty estimate. These uncertainties are at the level of 1% at $p_{T,ll} = 1\,\text{TeV}$ [64], and therefore would have an insignificant impact on our likelihood analysis used to derive the limits on the combinations $C_q/\Lambda^2$.

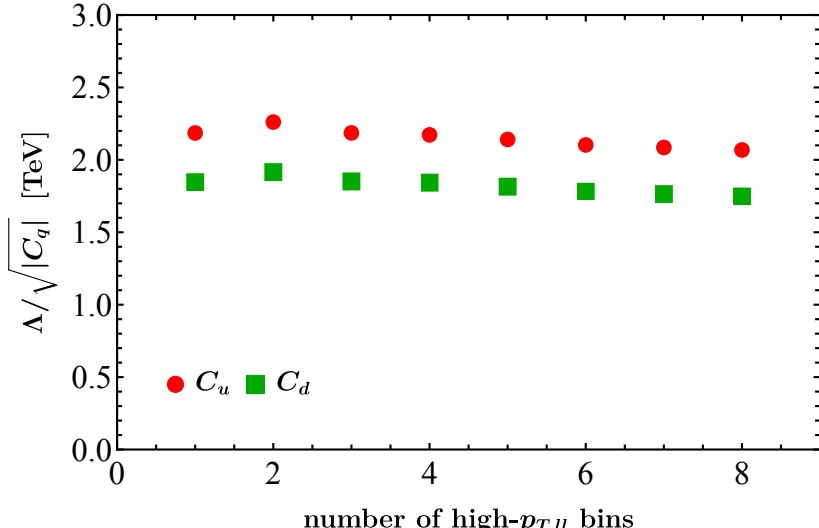

Figure 5: 95% CL lower limits on $\Lambda/\sqrt{|C_q|}$ are shown as a function of the number of high-$p_{T,ll}$ bins from the ATLAS measurement [1] included in the statistical analysis outlined in Section 4. The results for $C_u$ and $C_d$ are represented by red dots and green squares, respectively. Additional details are provided in the main text.

## B    Details on fit to $Z$+jet data

In this appendix, we present details on the Poisson likelihood analysis that has been used in Section 4 to derive limits on $C_q/\Lambda^2$ using the ATLAS data [1] on $Z$+jet production. Figure 5 illustrates the 95% CL lower limits on $\Lambda/\sqrt{|C_q|}$ as a function of the number of high-$p_{T,ll}$ bins from the ATLAS measurement included in our statistical analysis. The results for $C_u$ and $C_d$ are depicted by red dots and green squares, respectively. The first observation is that the derived lower bounds on $\Lambda/\sqrt{|C_q|}$ show little dependence on the number of high-$p_{T,ll}$ bins included in our Poisson likelihood analysis. The dependence of the limits on the number of bins can be qualitatively explained by noting that when only the highest bin is included in the fit, the limited statistical precision of the ATLAS measurement largely determines the resulting bound. Adding more high-$p_{T,ll}$ bins generally enhances the bounds on $\Lambda/\sqrt{|C_q|}$. However, the improvement is limited because the SM prediction remains consistently below the ATLAS data throughout the entire $p_{T,ll}$ range of $[200, 900]$ GeV, as illustrated in Figure 2. The values presented in (26) are derived from the two highest $p_{T,ll}$ bins, as this choice provides the most stringent upper limits on $|C_q|/\Lambda^2$. It is worth pointing out that our likelihood analysis also leads to lower bounds where $|C_q|/\Lambda^2 > 0$. However, these limits stem from the fact that the SM prediction for the normalized $p_{T,ll}$ spectrum is consistently lower than the ATLAS data, potentially due to unaccounted-for QCD, EW, or experimental effects, which could bias the results of the likelihood analysis. Therefore, we limited our fit to establishing upper limits on $|C_q|/\Lambda^2$, as these are less influenced by the discrepancy between theory and experiment compared to the lower bounds.

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
