# Peer review of "A tale of $Z$+jet: SMEFT effects and the Lam-Tung relation"

_SciPost Physics Core_

## Round 1 · Referee Report · Anonymous (Referee 1) · 2025-3-3

Strengths
1- The paper is clearly written. 2- Very careful analysis of uncertainties for the Z boson measurements. 3- Concise and easy-to-follow presentation of the results.
Weaknesses
1- There could be some discussion about other dimension-6 operators which can affect the same processes. 2-The robustness of the conclusions with respect to the assumptions made in the analysis could be discussed more.
Report
Requested changes
1- I would suggest that the authors discuss which other operators can enter in the processes they consider. Is there a possibility that effects of the dipole operators are masked by the presence of other operators in the EWPOs and in the $p_T$ spectrum but still visible in the angular coefficients?
2- The authors say that the linear terms in the Wilson coefficients are proportional to the Yukawa couplings. I think it is more precise to say the fermion masses rather than the Yukawa couplings.
3- I was wondering why a different Monte Carlo generation setup with different inputs was used for the transverse momentum spectrum and the extraction of the angular coefficients. Can the authors clarify?
4- The authors mention a larger value of the dipole coefficient which would result in effects 6 times larger than what they see. Does that also reproduce the right shape for the data? Could the authors perhaps plot it?
Recommendation
Ask for minor revision
We appreciate the referee's thorough review of our manuscript and their insightful comments. In this response, we have tried to address all their suggestions, making the necessary revisions to the manuscript. For clarity, we have included a version of the revised draft where all modifications and additions are highlighted in red.
Reply to the report of the referee:
(Q1) I would suggest that the authors discuss which other operators can enter in the processes they consider. Is there a possibility that effects of the dipole operators are masked by the presence of other operators in the EWPOs and in the $p_T$ spectrum but still visible in the angular coefficients?
(A1) As per the referee's recommendation, we have added a new Appendix C to the manuscript, providing a brief discussion of other types of Standard Model effective field theory (SMEFT) contributions to $Z$+jet production. Specifically, we examine a dimension-six operator that modifies the right-handed couplings between the $Z$ boson and up quarks. The discussion in Appendix C highlights that the deviation patterns in the $p_{T,ll}$ spectrum and the combination $A_0 - A_2$ of angular coefficients, induced by dimension-six operators affecting the $Z$-boson couplings to light quarks, differ from those generated by light-quark dipole operators, which are and remain the primary focus of our study. The findings in Appendix C strongly indicate that the $p_{T,ll}$ and $A_0 - A_2$ distributions can serve as useful tools for distinguishing dipole-type operators from effective interactions that alter the $Z$-boson couplings. Whether this also holds for dimension-six semileptonic operators or dimension-eight interactions is an interesting question, though we believe it lies beyond the scope of our study. We therefore do not try to address this question in the manuscript.
(Q2) The authors say that the linear terms in the Wilson coefficients are proportional to the Yukawa couplings. I think it is more precise to say the fermion masses rather than the Yukawa couplings.
(A2) We acknowledge the referee's point and have revised the manuscript accordingly.
(Q3) I was wondering why a different Monte Carlo generation setup with different inputs was used for the transverse momentum spectrum and the extraction of the angular coefficients. Can the authors clarify?
(A3) A key challenge in the computation of the angular coefficients $A_i$ is that evaluating the required expectation values involves averaging over oscillatory functions of $\theta$ and $\phi$. For example, the projector for $A_2$ includes terms like $\sin \theta$ and $\cos 2 \phi$. Additionally, calculating the combination $A_0 - A_2$ is complicated by numerical cancellations, both within the Standard Model (SM) and beyond. To overcome these difficulties, we utilized a private Drell-Yan code and applied adaptive sampling for the integral separately when computing $A_0$ and $A_2$. This approach proved significantly more efficient than using {\tt MadGraph5\_aMCNLO} to determine the angular coefficients $A_i$. Given the relative simplicity of computing the $p_{T,ll}$ distribution, we have directly used {\tt MadGraph5\_aMCNLO} to calculate this observable. We have included a short paragraph in the manuscript that addresses and explains this issue.
(Q4) The authors mention a larger value of the dipole coefficient which would result in effects 6 times larger than what they see. Does that also reproduce the right shape for the data? Could the authors perhaps plot it?
(A4) In response to the referee's comment, we have added a new Appendix D to the manuscript. In this new section of our work, we present results for the $p_{T,ll}$ spectrum and the $A_0 - A_2$ distribution corresponding to the Wilson coefficient choice for the up-quark dipole operator adopted in
@article{Li:2024iyj,
author = "Li, Xu and Yan, Bin and Yuan, C. -P.",
title = "{Lam-Tung relation breaking in $Z$ boson
production as a probe of SMEFT effects}",
eprint = "2405.04069",
archivePrefix = "arXiv",
primaryClass = "hep-ph",
reportNumber = "MSUHEP-24-006",
month = "5",
year = "2024"
}
which is the reference [22] in our article. The results presented in Appendix D demonstrate that the Wilson coefficient choice used in [22] leads to large enhancements in the tail of the $p_{T,ll}$ distribution, which are firmly ruled out by the data. Furthermore, the results for $A_0 - A_2$ show that incorporating the up-quark dipole contribution from [22] significantly impacts the Lam-Tung relation, helping to reduce the discrepancy between theory and the ATLAS measurement in the final $p_{T,ll}$ bin. However, it is important to note that even for $C_u/\Lambda^2= 1/(1.4 \, {\rm TeV})^2$, as chosen in [22], the beyond the SM result for $A_0 - A_2$ remains approximately $2\sigma$ ($2.5\sigma$) below the unregularized (regularized) data. These findings indicate that light-quark dipole operators are unlikely to be responsible for the excess observed in the $A_0 - A_2$ difference at high $p_{T,ll}$ in the ATLAS data.
We again thank the referee for their very useful feedback and hope that with the above explanations and the implemented changes the manuscript can be published in SciPost in its revised form.
Best regards,
Rhory Gauld, Ulrich Haisch, and Joachim Weiss
Attachment:
The authors have addressed all my comments in detail and I am happy to recommend publication.

Author: Ulrich Haisch on 2025-03-25 [id 5316]
(in reply to Report 2 on 2025-03-23)Reviewer 1 suggests publishing our work as it is, so no further action is needed.

---

## Editorial Decision

unknown